# OpenReview forum: "LoX: Low-Rank Extrapolation Robustifies LLM Safety Against Fine-tuning"
_colmweb.org/COLM/2025/Conference — COLM 2025_

### Official Review · Reviewer_a4V1 · 2025-05-11

**Rating:** 7
**Confidence:** 3
**Ethics Flag:** 1

**Summary:**

The paper addresses the vulnerability of safety-aligned LLMs to post-alignment fine-tuning attacks. It proposes LoX, a training-free method that extrapolates the top low-rank components of alignment weight updates to move the model toward a flatter and more robust safety region. The method reduces attack success rates substantially without degrading the model’s ability to adapt to new tasks. Experiments cover both LLaMA and Mistral models under several fine-tuning scenarios.

**Questions To Authors:**

Could you please provide your explanation for the comments in Reasons o Reject?

**Reasons To Accept:**

1. This is a training free method, which greatly enhances the applicability of this method.
2. Strong empirical results show consistent ASR reduction across multiple attack types and models.
3. Ablations and safety landscape visualizations support the core claims.

**Reasons To Reject:**

1. According to the align formula of this paper:
$\Delta W_{\text{align}} = W_{\text{align}} - W_{\text{base}}$.
 This method has an strong assumption: access to both aligned and unaligned model weights, which seems to be increasingly difficult to hold in the face of increasingly large and black-box LLM model deployments.
- Adding some newer and more capable baseline models—such as DeepSeek-R1-7b/14b[1]—could improve the robustness and generalizability of the evaluation.

2. Theoretical justification for why low-rank extrapolation improves safety is limited and mainly empirical.
- The entire approach is based on a core assumption that critical security-related information resides in low-rank subspaces of the model, and that fine-tuning erodes these specific subspaces.
- While the paper provides empirical evidence for this assumption, the theorical explanation behind the evidence seems to be lacking. Can some theoretical explanations be given for some smaller models?

## Reference
[1] *DeepSeek-R1: Incentivizing Reasoning Capability in LLMs via Reinforcement Learning*, arxiv

---

> ### Author Response · Authors · 2025-06-03
> **Response to Reviewer a4V1**
>
> We are grateful to the reviewer for recognizing the applicability of our method, as well as the results and ablation studies. Below, we provide detailed responses to each of your questions.
>
>  > According to the align formula of this paper: delta = align - base. This method has an strong assumption: access to both aligned and unaligned model weights, which seems to be increasingly difficult to hold in the face of increasingly large and black-box LLM model deployments.
>
> We thank the reviewer for raising this concern. We would like to emphasize that our assumption applies to scenarios where the person performing the safety alignment is also interested in ensuring its robustness. For example, the vendor of the aligned models. In such cases, LoX can be effectively used to augment the alignment.
>
> > Adding some newer and more capable baseline models—such as DeepSeek-R1-7b/14b[1]—could improve the robustness and generalizability of the evaluation.
>
> We appreciate the reviewer’s insightful suggestion. In response, we have aligned DeepSeek-R1-Distill-Qwen-1.5B using the same methodology described in our paper. The resulting model already demonstrates strong robustness, with ASR values of 0.02 on GSM8K and Alpaca, and 0.01 under the Identity Shifting Attack. Given this high level of robustness, additional robustness is unnecessary in this case, as essentially no safety degradation is observed. We would like to emphasize that our primary contribution lies in addressing safety degradation when it does arise.
>
>  >Theoretical justification for why low-rank extrapolation improves safety is limited and mainly empirical. The entire approach is based on a core assumption that critical security-related information resides in low-rank subspaces of the model, and that fine-tuning erodes these specific subspaces. While the paper provides empirical evidence for this assumption, the theorical explanation behind the evidence seems to be lacking. Can some theoretical explanations be given for some smaller models?
>
> We thank the reviewer for this suggestion. We agree that this could be a limitation for an empirical research. But we believe our empirical contributions are sound and novel as reviewers agree. We will leave this for future work.

---

> > ### Comment · Reviewer_a4V1 · 2025-06-04
> >
> > Thanks for the response and it answered most of my questions. I'm willing to increase my score based on that.

---

> > > ### Author Response · Authors · 2025-06-08
> > >
> > > Thank you for your thoughtful follow-up and for considering our responses. We're glad we could address most of your questions, and we truly appreciate your willingness to reconsider your score.

---

### Official Review · Reviewer_NrFf · 2025-05-11

**Rating:** 7
**Confidence:** 3
**Ethics Flag:** 1

**Summary:**

The manuscript proposes a new training-free alignment technique, LoX, to improve the robustness of aligned models against fine-tuning attacks (both intentional and unintentional). I quite enjoyed reading the paper, particularly the sections that explain the underlying intuition of the technique. I believe the community will benefit from this work. However, one important question remains unanswered: What if adversaries deliberately set the alpha parameter to a negative value to intentionally degrade model alignment?

**Questions To Authors:**

- In the Ethics section, please clarify how the risk of misuse (e.g., setting alpha to negative values) can be mitigated.
- Regarding the statement: “As k increases, changes in Rft/Ralign become less significant, highlighting the importance of the top-ranks on the safety robustness” — could you elaborate on why the top ranks are particularly important here?
- Figure 2 is somewhat difficult to read due to small marker sizes. Could you adjust the graph to improve readability?
- Nit: Line 241 — typo: “Iddentity Shifting” → should be “Identity Shifting”.

**Reasons To Accept:**

- The paper is well-written and easy to follow.
- The proposed technique is simple (training-free) yet demonstrates strong performance.
- Sections 3.3 and 5.3 provide valuable insights into how the technique works, fostering a deeper understanding of alignment post-training.
- The method offers an efficient way for model developers to enhance the robustness of aligned models without significantly compromising utility.

**Reasons To Reject:**

A potential risk is that bad actors might exploit the technique to reduce model alignment. Specifically, since the alpha parameter can be set to a negative value, it could be used to amplify safety degradation.

---

> ### Author Response · Authors · 2025-06-03
> **Response to Reviewer NrFf**
>
> We are grateful to the reviewer for acknowledging the quality of our writing, as well as the simplicity, efficacy, and thorough analysis of our technique. Below, we provide detailed responses to each of your questions.
>
> > A potential risk is that bad actors might exploit the technique to reduce model alignment. Specifically, since the alpha parameter can be set to a negative value, it could be used to amplify safety degradation.
>
> > In the Ethics section, please clarify how the risk of misuse (e.g., setting alpha to negative values) can be mitigated.
>
> We thank the reviewer for raising this concern. We argue that, for our method to work, it requires both an aligned (safe) and unaligned (unsafe) checkpoints. If the individual can apply LoX, it implies that they have access to the unsafe model. Therefore, the individual would have no motivation to diminish the alignment. We will update the ethics section to address this discussion.
>
> > Regarding the statement: “As k increases, changes in Rft/Ralign become less significant, highlighting the importance of the top-ranks on the safety robustness” — could you elaborate on why the top ranks are particularly important here?
>
> In this experiment, we increase the samples in fine-tuning (marker size), which results in more safety degradation (y-axis). We hypothesize that the safety degradation is a result of the change of the ratio R_ft / R_align​ after fine-tuning. Therefore, we test if the safety degradation is correlated with the ratio R_ft / R_align​.  In the figure, the correlation is more statistically significant when the rank k is smaller (i.e., top-rank subspace are measured by the ratio). In contrast, with larger rank k, the ratio is barely changed by the safety degradation. In other words, the top-rank subspace (the smaller rank) plays a more important role, impacting safety during fine-tuning, than the full-rank subspace.
>
> > Figure 2 is somewhat difficult to read due to small marker sizes. Could you adjust the graph to improve readability?
>
> We appreciate the feedback regarding the readability of Figure 2 and will improve the marker sizes in the next revision of the paper.
>
> > Nit: Line 241 — typo: “Iddentity Shifting” → should be “Identity Shifting”.
>
> We thank the reviewer for pointing out the typo and will correct it in the next revision.

---

> > ### Author Response · Authors · 2025-06-08
> >
> > We sincerely appreciate the reviewers for their thoughtful feedback and constructive comments, which have been instrumental in enhancing our paper. As a gentle reminder following our rebuttal, the discussion period is nearing its end. We hope to make the most of this opportunity to engage further, answer any additional questions, and continue improving our work. Have you had a chance to review our responses below? We did our best to address each of your concerns thoroughly and clearly. Please let us know if our replies were satisfactory, and we’re more than happy to provide further clarification or details if needed.

---

### Official Review · Reviewer_uZoX · 2025-05-13

**Rating:** 7
**Confidence:** 3
**Ethics Flag:** 1

**Summary:**

In this work, the adversarial robustness of LLM model fine-tuning are discussed. A novel yet efficient method called Low-Rank-Extrapolation (LoX) is proposed to robustify the model during finetuning. The key hypothesis of this work is that the model will lose it's safety-related knowledge during finetuning, demonstrated by the difference between SVD decomposition of the  $\Delta$ during model update step. The proposed method is to add an additional regularization term to keep the alignment(safety) knowledge during model finetuning stage. It's evaluated on multiple adversarial attack datasets.

**Questions To Authors:**

See the above section

**Reasons To Accept:**

- The paper is well-written and the proposed method is neat
- The proposed method achieved significant improvement in preventing safety regression of LLM after finetune.
- It's a novel idea to study the safety-knowledge preserving property during LM training.

**Reasons To Reject:**

- “ fine-tuning can unintentionally alter low-rank subspaces of model weights related to safety of well-aligned LLMs, and finally result in diminished safety”. This is a pretty strong claim & hypothesis, it would be great if there are some theoretical analysis that can be conducted as evidence. For example, why finetuning only degradate the safety knowledge, how do we split the knowledge space of LLM?

- Lack some reference about the adversarial robustness of LMs, like [1, 2, 3]. Would be good to discuss and compare with those methods.

[1] Dong, Xinshuai, et al. "How should pre-trained language models be fine-tuned towards adversarial robustness?." Advances in Neural Information Processing Systems 34 (2021): 4356-4369.

[2] Hou, Bairu, et al. "TextGrad: Advancing Robustness Evaluation in NLP by Gradient-Driven Optimization." The Eleventh International Conference on Learning Representations.

[3] Kim, Jaehyung, et al. "RoAST: Robustifying Language Models via Adversarial Perturbation with Selective Training." Findings of the Association for Computational Linguistics: EMNLP 2023. 2023.

---

> ### Author Response · Authors · 2025-06-03
> **Response to Reviewer uZoX**
>
> We thank the reviewer for recognizing the clarity of our writing, the effectiveness of our method, and the novelty of studying safety preservation in language model training.
>
> > “ fine-tuning can unintentionally alter low-rank subspaces of model weights related to safety of well-aligned LLMs, and finally result in diminished safety”. This is a pretty strong claim & hypothesis, it would be great if there are some theoretical analysis that can be conducted as evidence. For example, why finetuning only degradate the safety knowledge, how do we split the knowledge space of LLM?
>
> First, we would like to clarify that we didn’t claim that fine-tuning affects only safety, as catastrophic forgetting is a well known problem in the ML community. If the reviewer could point to the part in the text that makes them understand that, we would be happy to edit it.
>
> Regarding the theoretical analysis, we thank the reviewer for the suggestion. While we do not yet have strong theoretical insights, we believe our empirical contributions—specifically, the new insights on the safety subspace and our method for robustifying safety-aligned models—are both sound and novel, as acknowledged by the reviewers. We plan to explore the theoretical aspects in future work.
>
> > Lack some reference about the adversarial robustness of LMs, like [1, 2, 3]. Would be good to discuss and compare with those methods.
>
> We thank the reviewer for sharing this related work. We provide a discussion comparing our work with theirs below. This discussion will be integrated in our final revision.
>
> RIFT [1]:
> RIFT addresses the problem of catastrophic forgetting during adversarial fine-tuning by encouraging pre-trained language models to retain their robust, generic linguistic features. RIFT could be categorized as a fine-tuning stage defense, as it changes fine-tuning dynamics to retain information from the base model. Therefore, RIFT is not suitable for the scenario where a harmful attacker (i. e. a person with intention of jailbreaking the model through fine-tuning) has control over the fine-tuning process. On the other hand, LoX can be applied after alignment stage, being suitable for such scenario.
>
>
> Textgrad [2]:
> TextGrad  adresses the lack of a principled, gradient-based method for generating fluent adversarial examples in NLP, enabling more effective robustness evaluation and adversarial training. Their work focuses on classification tasks, and not text generation like ours.
>
> RoAST [3]:
> ROAST tackles the lack of unified robustness in fine-tuned language models by combining adversarial perturbations with selective parameter updates, improving resilience across multiple robustness dimensions. Similar to RIFT, RoAST can be categorized as a fine-tuning stage defense, and therefore is also not suitable for the scenario described.

---

> > ### Comment · Reviewer_uZoX · 2025-06-03
> > **Thanks for the comments**
> >
> > - "We hypothesize that fine-tuning can unintentionally alter low-rank subspaces of model, weights related to safety of well-aligned LLMs, and finally result in diminished safety". This is from row 149-150. It reads like the finetuning will only impact weights related to safety.
> > - Discussion about related work. I think LoX here can also be categorized as a special finetuning method. So it's not that principal to classify the methods into "before" and "after" finetuning stage

---

> > > ### Author Response · Authors · 2025-06-06
> > >
> > > Thank you for your follow-up questions, which make the previous discussion clearer. We are trying our best to add follow-up responses, hopefully to address your concerns.
> > >
> > > > "We hypothesize that fine-tuning can unintentionally alter low-rank subspaces of model, weights related to safety of well-aligned LLMs, and finally result in diminished safety". This is from row 149-150. It reads like the finetuning will only impact weights related to safety.
> > >
> > > We thank the reviewer’s clarification on the question. We admit that the original statement could be misleading. It is true that finetuning will NOT only impact weights related to safety. We rephrase the statement as the following one and will integrate into our next revision.
> > >
> > > "We hypothesize that fine-tuning may inadvertently alter various parameters, including safety-sensitive ones, and when these safety-sensitive low-rank subspaces are affected, model safety can be significantly degraded."
> > >
> > >
> > > > Discussion about related work. I think LoX here can also be categorized as a special finetuning method. So it's not that principal to classify the methods into "before" and "after" finetuning stage
> > >
> > > We sincerely thank the reviewer for this thoughtful comment. We agree that, in a broad sense, LoX could be viewed as related to fine-tuning methods, since it can be applied in conjunction with the fine-tuning process. However, we see an important distinction: unlike conventional fine-tuning, LoX does not involve additional data or modify the optimization procedure. Its simplicity and minimal intervention set it apart from typical fine-tuning-based approaches.
> > >
> > > This distinction becomes especially important in adversarial settings, where the fine-tuning process may be entirely controlled by an attacker. In such cases, defenses that rely on modifying the fine-tuning stage are less practical, as the attacker has no incentive to incorporate them. In contrast, LoX is applied prior to any downstream fine-tuning, enabling it to remain effective even when the subsequent fine-tuning is untrusted or adversarial.

---

> > > ### Author Response · Authors · 2025-06-08
> > >
> > > Thank you for taking the time to review our paper and share your insightful comments and suggestions. We were wondering if you’ve had a chance to look over our responses to your questions? We’d be happy to offer further explanations or address any additional questions you may have.

---

> > > > ### Comment · Reviewer_uZoX · 2025-06-09
> > > > **Thanks for the response**
> > > >
> > > > The explanation makes sense to me, I will keep my current rating

---

### Official Review · Reviewer_3vEX · 2025-05-13

**Rating:** 7
**Confidence:** 3
**Ethics Flag:** 1

**Summary:**

This paper investigates how to mitigate the degradation of safety protections from alignment training that often occurs during further fine-tuning, even on benign datasets. The authors observe that the alignment update $\Delta W_{\text {align}}$ is effectively low-rank and that fine-tuning vanishes the sub-space. They propose a novel method, LoX, simply pre-adds a low-rank projection of $\alpha \Delta W_{\text {align}}$ before fine-tuning. Experiments show that LoX substantially improves attack-success rates while preserving task accuracy and helpfulness.

**Questions To Authors:**

1. The safety landscape in section 5.3 is interesting. Do you have a hypothesis why adding this low-rank approximation of $\Delta W_{\text {align}}$ help models move to a flatter region?
2. I am curious about the model after applying LoX and before training. What are the behavior changes?

**Reasons To Accept:**

1. Figure 2 on the relationship between $R_{\mathrm{ft}} / R_{\mathrm{align}}$ is insightful and well-articulated.
2. The proposed method is data-free and post-hoc, making it applicable in many scenarios.
3. Comprehensive ablation studies. I like how LoX compares to full-rank, showing the benefits or ignoring less important directions.

**Reasons To Reject:**

1. I would appreciate a more thorough discussion on why adding the full $\Delta W_{\text {align}}$ makes training unstable. What are the performance difference?

---

> ### Author Response · Authors · 2025-06-03
> **Response to Reviewer 3vEX**
>
> We thank the reviewer for recognizing the applicability of our method and the studies conducted in our work. Please find our point-by-point responses to the questions below.
>
> > I would appreciate a more thorough discussion on why adding the full rank makes training unstable. What are the performance difference?
>
> We appreciate your request for more discussion. But to our best knowledge, we did not claim that adding full-rank will make training unstable.  Instead, we would like to clarify that adding full-rank components degrades the model prior to the fine-tuning attack; we do not claim any effect on fine-tuning dynamics. As stated in line 352, we hypothesize that this degradation is due to excessive noise in the bottom ranks. The impact of adding full-rank components compared to low-rank ones is illustrated in Figure 8 (model output examples) and Table 7 (utility metric comparison) in Appendix E. We will revise the main text to explicitly reference these results.
>
> > The safety landscape in section 5.3 is interesting. Do you have a hypothesis why adding this low-rank approximation of delta help models move to a flatter region?
>
> Prior work [1] has shown that the safety landscape exhibits a basin shape, and that fine-tuning can drag the models that are close to the border out of the basin. We conjecture that alignment stops close to the border of the basin (due to gradient flattening), and that extrapolating in the safety direction (i. e. adding delta to the model) pushes the model deeper into the basin. The low-rank approximation performs better because it avoids extrapolating the noise present in the bottom ranks, allowing us to move “further” into the safe region without degrading the model’s outputs.
>
> > I am curious about the model after applying LoX and before training. What are the behavior changes?
>
> As previously mentioned in the answer to the first question, Figure 8 and Table 7 in Appendix E show the effects of applying LoX prior to the fine-tuning attack. We also provide additional results comparing utility metrics for the base model (baseline), LoX, and the full-rank variant (ExPO), all evaluated with an extrapolation factor of α = 1.25.
>
> | Model  | Method | Helpfulness | Clarity | Factuality | Depth | Engagement |
> | :---: | :---: | :---: | :---: | :---: | :---: | :---: |
> | Llama-2 22.5k | Baseline | 1.36 | 1.50 | 1.71 | 1.27 | 1.30 |
> | Llama-2 22.5k | ExPO | 1.18 | 1.37 | 1.39 | 1.14 | 1.25 |
> | Llama-2 22.5k | LoX | 1.18 | 1.42 | 1.55 | 1.15 | 1.30 |
> | Llama-2 65.6k | Baseline | 2.08 | 2.16 | 2.50 | 1.77 | 1.81 |
> | Llama-2 65.6k | ExPO | 1.53 | 1.80 | 1.93 | 1.35 | 1.62 |
> | Llama-2 65.6k | LoX | 1.79 | 2.09 | 2.33 | 1.51 | 1.83 |
>
>
> The results show a degradation across all five metrics when comparing both LoX and ExPO to the base model, with LoX mitigating this degradation relative to ExPO. This indicates that using only top-ranks is capable of preserving more the original performance of the model, when compared to full-rank.
>
> References:
> [1] Peng, S., Chen, P.Y., Hull, M., & Chau, D. (2024). Navigating the Safety Landscape: Measuring Risks in Finetuning Large Language Models. In Advances in Neural Information Processing Systems (pp. 95692–95715). Curran Associates, Inc..

---

> > ### Comment · Reviewer_3vEX · 2025-06-05
> >
> > Dear authors, thank you for your response. I will stick to my scores since it's already leaning acceptance.

---

> > > ### Author Response · Authors · 2025-06-08
> > >
> > > Thank you for your reply and for taking the time to review our work. We truly appreciate your thoughtful evaluation and support.

---

### Decision · Program_Chairs · 2025-07-08

**Decision:**

Accept

**Comment:**

This paper presents LoX, a novel and efficient method to mitigate the degradation of safety protections in LLMs during post-alignment fine-tuning, even on benign data. The motivation is based on the empirical observation that alignment updates tend to be low-rank and can be easily erased through additional fine-tuning. LoX mitigates this degradation by pre-applying a low-rank projection to preserve alignment-relevant subspaces.

The paper is well-written, clearly motivated, and proposes a new and practical solution to an important problem in LLM robustness—namely, the post-alignment safety degradation after further fine-tuning. All three reviewers praised the clarity of presentation, novelty, and strong empirical results.

In summary, here are the pros of the paper:
- Addresses a timely and practically relevant challenge in LLM deployment.
- Simple, post-hoc, and data-free technique with minimal computational overhead.
- Strong empirical results with helpful visual explanations and ablations.
- Well-organized and clearly written.

However, the paper can still be improved in the following aspects:
- Lacks theoretical grounding for key assumptions about the nature of safety knowledge.
- Missing discussion of and comparison to some related work on robustness.
- Potential misuse of the method for adversarial purposes is acknowledged but not addressed in depth.